# Measurement of Cardiac-Specific Biomarkers in the Emergency Department: New Insight in Risk Evaluation

**DOI:** 10.3390/ijms242115998

**Published:** 2023-11-06

**Authors:** Nadia Aspromonte, Martina Zaninotto, Alberto Aimo, Isabella Fumarulo, Mario Plebani, Aldo Clerico

**Affiliations:** 1Department of Cardiovascular and Thoracic Sciences, Catholic University of the Sacred Heart, 00168 Rome, Italy; nadia.aspromonte@policlinicogemelli.it (N.A.); isabella.fumarulo01@icatt.it (I.F.); 2Department of Cardiovascular and Thoracic Sciences, A. Gemelli University Policlinic Foundation IRCCS, 00168 Rome, Italy; 3Department of Laboratory Medicine, University-Hospital of Padova, 35129 Padova, Italy; martina.zaninotto@sanita.padova.it; 4CNR Foundation—Regione Toscana G. Monasterio, 56127 Pisa, Italy; alberto.aimo@santannapisa.it; 5Department of Medicine-DIMED, University of Padova, 35129 Padova, Italy; mario.plebani@unipd.it; 6Coordinator of the Study Group on Cardiac Biomarkers of the Italian Societies of Laboratory Medicine, 56127 Pisa, Italy

**Keywords:** emergency department, cardiac biomarkers, acute cardiac disease

## Abstract

The aim of this article review is to analyze some models and clinical issues related to the implementation of accelerated diagnostic protocols based on specific cardiac biomarkers in patients admitted to the emergency department (ED) with symptoms compatible with acute cardiac disorders. Four specific clinical issues will be discussed in detail: (a) pathophysiological and clinical interpretations of circulating hs-cTnI and hs-cTnT levels; (b) the clinical relevance and estimation of the biological variation of biomarkers in patients admitted to the ED with acute and severe diseases; (c) the role and advantages of the point-of-care testing (POCT) methods for cardiac-specific biomarkers in pre-hospital and hospital clinical practice; and (d) the clinical role of specific cardiac biomarkers in patients with acute heart failure (AHF). In order to balance the risk between a hasty discharge versus the potential harms caused by a cardiac assessment in patients admitted to the ED with suspected acute cardiovascular disease, the measurement of specific cardiac biomarkers is essential for the early identification of the presence of myocardial dysfunction and/or injury and to significantly reduce the length and costs of hospitalization. Moreover, specific cardiac biomarkers (especially hs-cTnI and hs-cTnT) are useful predictors of mortality and major adverse cardiovascular events (MACE) in patients admitted to the ED with suspected acute cardiovascular disease. To guide the implementation of the most rapid algorithms for the diagnosis of Non-ST-Elevation Myocardial Infarction (NSTEMI) into routine clinical practice, clinical scientific societies and laboratory medicine societies should promote collaborative studies specifically designed for the evaluation of the analytical performance and, especially, the cost/benefit ratio resulting from the use of these clinical protocols and POCT methods in the ED clinical practice.

## 1. Introduction

In September 2021, a systematic review by the Agency for Healthcare Research and Quality U.S. Department of Health and Human Services (AURQ Report) showed that (a) overall diagnostic accuracy in the emergency department (ED) is high, but some patients receive an incorrect diagnosis (~5.7%); (b) about 2.0% of these patients suffer an adverse event because of the incorrect diagnosis; and (c) about 3.0% of adverse events are serious (~0.3%) [1]. This document identified the 15 clinical conditions associated with the most serious harm in case of misdiagnosis, accounting for 68% of all cases of serious harm, in the ED [1]. Among these conditions, cardiovascular disorders are represented by stroke (position 1), myocardial infarction (position 2), aortic aneurysm and dissection (position 3), and cardiac arrhythmias (position 12). Solutions to enhance bedside diagnostic processes are needed, and these should target the most commonly misdiagnosed clinical presentations, leading to serious harms [1].

In particular, heart failure or myocardial damage significantly increase the risk of death in patients admitted to the ED with acute non-cardiac conditions such as sepsis, pneumonia, and thromboembolism [2,3,4,5,6,7]. In particular, high levels of values of cardiac troponin I (cTnI) and T (cTnT), revealing the presence of an acute myocardial injury [8], were frequently found in patients with COVID-19-related pneumonia and/or sepsis and are associated with adverse outcomes such as cardiac failure, arrhythmias, and death [6,7].

In December 2022, a multi-organizational document [9] published by the American College of Emergency Physicians (ACEP) and other American societies related to emergency medicine, stressed an important problem in ED care. The main concern is that ED physicians prioritize stabilization of critically ill patients over the identification of life-threatening conditions [9]. Indeed, due to the constraints related to ED activity, physicians often make only a preliminary, working diagnosis, trusting that their colleagues in clinical wards will have more time and easier access to more complex investigational procedures in order to acquire more detailed information to make the final diagnosis [9]. On the other hand, patients admitted to the ED represent the most diagnostically challenging people. Therefore, more resources should be allocated to increase the speed and accuracy of initial diagnosis and treatment [9].

Every year, millions of patients are admitted to the EDs of developed countries with clinical symptoms related to acute coronary syndrome (ACS), but the diagnosis of ACS is confirmed in less than 10% of them [10,11,12,13]. In the last years, several documents and international guidelines have been published to suggest some accelerated diagnostic protocols concerning the differential diagnosis of ACS and a more accurate risk stratification of patients admitted to the ED with suspected ACS [14,15,16,17,18,19,20,21,22,23]. 

In 2021, Hendley et al. [10] properly observed that an ethical dilemma often arises with patients admitted to the ED with thoracic pain, when clinicians try to balance the risk of not diagnosing an ACS with the potential harm caused by a more accurate cardiac work-up [15,17,20,21,22,23].

For ten years now, high-sensitivity immunoassays for cardiac troponin I (hs-cTnI) and cardiac troponin T (hs-cTnT) have been recommended by all the international guidelines as gold standard laboratory methods for the detection of myocardial injury and diagnosis of acute myocardial infarction (AMI) [8,14,15,16,19,24]. In particular, the Fourth Universal Definition of Myocardial Infarction [8] defines myocardial injury as a distinct condition characterized by at least one hs-cTnI or hs-cTnT value above the 99th percentile of the biomarker distribution values, assessed in a healthy adult reference population (99th upper reference limit—URL—value). 

The use of hs-cTnI and hs-cTnT methods has allowed a progressive reduction in the time to diagnosis of NSTEMI (non-ST-elevation myocardial infarction) from 6 to 12 h to less than 3 h in patients admitted to the ED [14,15,16,17,18,19]. The 2020 European Society of Cardiology (ESC) guidelines recommend the fastest clinical algorithms with blood sampling on admission and after 1 or 2 h (0–1 h or 0–2 h) [15]. This recommendation is based on data suggesting that these algorithms (especially the 0–1 h algorithm) allow to rule in and rule out NSTEMI in the shortest possible time [15]. Furthermore, in the last 5 years, the clinical results of some point-of-care-testing (POCT) methods (i.e., hs-cTnI POCT), with analytical performance equal to hs-cTnI methods, have been developed and evaluated [25,26,27,28,29,30,31,32,33,34,35,36].

Considering these analytical improvements and clinical evidence [14,15,16,17,18,19,25,26,27,28,29,30,31,32,33,34,35,36], recent documents and guidelines [14,15,16,17,36,37] recommend the use of accelerated diagnostic protocols, based on the use of hs-cTn, in order to reduce the length of stay in the ED and hospital and thus reducing costs related to the management of patients with thoracic pain, ACS, or acute heart failure (AHF).

## 2. Article Aim and Structure

In this review article, we will analyze the models and clinical issues related to the implementation of accelerated diagnostic protocols in pre-hospital and hospital clinical practice of assay for specific cardiac biomarkers in patients admitted to the ED with symptoms compatible with acute cardiac disorders. Four specific clinical issues will be discussed in detail: (a) pathophysiological and clinical interpretations of circulating hs-cTnI and hs-cTnT levels; (b) the clinical relevance and estimation of the biological variation in biomarkers in patients admitted to the ED with acute and severe diseases; (c) the role and advantages of POCT methods for cardiac specific biomarkers in pre-hospital and hospital clinical practice; and (d) the clinical role of specific cardiac biomarkers in patients with AHF. Clinical scientific societies and laboratory medicine societies should promote collaborative studies specifically designed for the evaluation of the analytical performance and, especially, the cost/benefit ratio resulting from the use of the most rapid algorithms for diagnosis of NSTEMI and the use of POCT methods, in the specific clinical conditions related to ED clinical practice.

## 3. Cardiac Biomarkers in the Setting of ED

### 3.1. Clinical Interpretations of Circulating hs-cTnI and hs-cTnT Levels

All recent international guidelines agree that hs-cTnI and hs-cTnT are the first-line laboratory biomarkers for the diagnosis of myocardial injury and AMI [8,14,15,16,19,24]. Two fundamental analytical criteria for the hs-cTnI and hs-cTnT assays are required: (1) the coefficient of variation should be ≤10% at the 99th percentile URL value; and (2) troponin should be measurable in ≥50% of a reference population with at least 600 individuals of both sexes including at least 300 healthy women with biomarker concentrations above the limit of detection (LoD) of the method [24]. According to these quality specifications [24], the accurate measurement of circulating hs-cTnI and hs-cTnT levels is challenging due to biomarker concentration in healthy women, which is lower than in healthy men of the same age [38,39,40,41]. Indeed, the 10% of biomarkers values, measured with the most popular hs-cTnI methods in a large Italian reference population, are actually ≤2 ng/L (i.e., on average the LoD values of the methods) [40]. Considering the hs-cTnT method, about 20–25% of apparently healthy adult European and Chinese men and women had cTnT values ≤3 ng/L (i.e., the LoD of the method) [41,42,43]. 

Circulating levels of hs-cTnI and hs-cTnT in healthy adult subjects of both sexes should be considered a reliable index of the physiological cardiomyocyte renewal [38,39,40,44,45,46,47,48]. In particular, several clinical studies have reported that the 99th percentile URL values of hs-cTn methods range on average from 13 to 47 ng/L, corresponding to the renewal of about 30–40 mg of the myocardium [26,38,39,40,41,42,43,44,45,46]. Accordingly, the mean biomarker concentrations of about 3–5 ng/L (typical of adult healthy subjects) are related to a myocardial volume ≤10 mg. As the amount of daily renewal of myocardial tissue is too low to be detected by cardiac imaging techniques, such as magnetic resonance or positron emission tomography [46,47], the Fourth Universal Definition of Myocardial Infarction has defined myocardial injury considering the 99th percentile URL value of hs-cTnI and hs-cTnT assays as the cut-off value [8].

### 3.2. Clinical Relevance of the Biological Variation in Cardiac Biomarkers

The Fourth Universal Definition of Myocardial Infarction [8] is based on the clinical distinction between two different clinical conditions: (a) acute myocardial injury, characterized by significant changes over time in the hs-cTnI and hs-cTnT levels in a patient, and (b) chronic myocardial injury (i.e., heart damage), characterized by nearly stable biomarker levels. Accordingly, the biological variation in circulating levels of biomarkers, and especially how this variation is evaluated, are fundamental issues related to the routine clinical work up of patients admitted to the ED for chest pain or ACS [38,48,49,50,51].

hs-cTnI and hs-cTnT have a very low index of intra-individual biological variation (i.e., individual index, II), on average 0.3, like creatinine. For comparison, the individual index value of cardiac natriuretic peptides (NPs), including BNP and NT-proBNP, is >0.6 [48,49,50,51]. Biomarkers with an II value < 0.6 usually show a better correlation with some fundamental individual physiological characteristics (in particular sex, age, body height, and muscular mass) [48], and these laboratory tests are preferable according to the principles of personalized and precision medicine. To optimize the diagnostic accuracy of the hs-cTnI and hs-cTnT methods, changes in biomarker concentrations should be checked on serial samples in a single patient using the same assay method, rather than comparing a single value with reference values estimated in a reference population (such as the 99th percentile URL or a clinical cut-off value) which are characterized by a large confidence interval [48,49,50,51].

The latest guidelines recommend specific algorithms to evaluate the changes over time in hs-cTnI and hs-cTnT in patients with thoracic pain [8,14,15,16,17,18,19,20]. In particular, the 2020 ESC guidelines [15] recommend the most rapid 0 h/1 h algorithm as the best option, considering the first sample draw at 0 h (i.e., immediately on admission to the ED) and then the second after 1 h. Alternatively, the guidelines recommend the 0 h/2 h algorithm considering the first sample at admission and then the second one after 2 h [15]. Moreover, changing patterns over time (indicated as deltas or with the Greek symbol Δ) should be evaluated by taking into consideration the hs-cTnI and hs-cTnT concentrations measured in the two serial samples of the patient according to the algorithm (i.e., absolute change) [15]. 

The 2020 ESC guidelines state that the safety (as quantified by the negative predictive value, NPV) and sensitivity are very high (>99%), including in the subgroup of patients presenting very early (i.e., <2 h from symptom onset) [15]. Moreover, the rapid algorithms (especially the fastest 0 h/1 h algorithm) should be preferred because they substantially reduce the delay to diagnosis, producing shorter patient stays in the ED compared to the 0 h/3 h recommend by the previous 2015 ESC guidelines [52]. Finally, the 2020 ESC guidelines also recommend specific levels for rule-in and rule-out at time 0, using only one admission value. The method-specific cut-off levels indicated by the 2020 ESC guidelines [15] for the rule-out at time 0 are usually very similar to the LoD value of the hs-cTnI and hs-cTnT methods, showing an NPV ≥ 99% [14,18,48,49,50]. Conversely, the method-specific cut-off levels indicated by the 2020 ESC guidelines for the rule-in at time 0 have a slightly lower diagnostic accuracy, with an optimal threshold for rule-in selected to allow for a positive predictive value (PPV) of about 70% [15], so these patients always require further non-invasive and/or invasive investigations to confirm the diagnosis of MI [14].

In accordance with some of the recent documents and guidelines [14,16,18,50], there are still some additional clinical issues to consider [15]. First, reliable evidence for the delta values for the fastest algorithms for some hs-cTnI methods is still lacking or limited [14,16,18,51]. Secondly, it is always necessary to ensure that the symptoms have begun at least 3 h before the collection, because hs-cTnI and hs-cTnT levels increase slowly in the first hours; therefore, it is hard to detect a significant variation in the biomarker levels too early [15,18,19]. Third, the cut-off and delta values recommended by ESC 2020 are not sex-specific. It is well known that the 99th percentile URL values are significantly higher in men compared to women, thus this difference should be taken into consideration in the diagnosis of myocardial injury and AMI, using specific cut-off values for method and sex [8,14,16,18,19,39,40,41,49,53,54,55,56,57,58]. Indeed, non-sex-specific cut-off values can lead to an underestimation of the diagnosis of myocardial injury and AMI in women admitted to the ED with thoracic pain or ACS, especially when the hs-cTnI methods are used [58,59,60,61,62,63,64,65]. 

Recent documents supported by the Italian Societies of Laboratory Medicine [18,49,51] have suggested an alternative and easier method to accurately evaluate biomarker variations in patients admitted to the ED for thoracic pain or ACS, compared to the ESC 2020 recommendations [15]. It seems much easier to estimate the variation in hs-cTnI and hs-cTnT concentrations as a percentage rather than as absolute changes as recommend by ESC guidelines, by measuring the relative change values (RCV) [48,49,50,51]. This practical approach is based on the evidence that the imprecision profiles of the hs-cTnI and hs-cTnT are not significantly different [18,40,48,49,50,51,66,67,68,69]. Indeed, the RCV values for a series of two samples were reported to vary on average by 32%, in agreement with many experimental and clinical studies including both healthy adult subjects and patients admitted to the ED in whom the presence of acute heart damage was excluded [18,40,48,49,50,51,65,66,67,68,69,70,71]. The percent change rule (RCV%) for evaluating the kinetics of hs-cTnI and hs-cTnT values in patients admitted to the ED with suspected MI is recommended by many guidelines and expert papers [8,14,16,17,19,57].

The evaluation of the kinetics of hs-cTnI and hs-cTnT values as RCV% has the additional advantage of providing a more accurate estimate of cardiovascular risk for major adverse cardiovascular events (MACE) or death, even in patients presenting with chronic or acute non-ischemic myocardial damage, as recommend by some recent guidelines [17,19,53].

### 3.3. Role and Advantages of POCT Methods for Cardiac Troponin in Clinical Practice

The very recent development of some POCT methods for cTnI with high analytical sensitivity (POCT hs-cTnI methods) represents fundamental progress because these methods can significantly reduce the turnaround time (TAT) of biomarker measurement in patients with NSTEMI [25,26,36,37]. Furthermore, the POCT methods can be used at home, in the outpatient clinic, or in the ambulance in order to reduce unnecessary transfer to hospital and allow patient management in rural general practice [34,35,72] (Figure 1).

Form an analytical point of view, chemiluminescence and fluorescence techniques are used interchangeably in POCT hs-cTnI methods (Table 1) [26,49]. Both these techniques give off a photon as an electron relaxes from a higher energy state to a lower energy state, but the difference lies in the method used to excite that electron to a higher energy state in the first place. In fluorescence, the electron is kicked up to a higher energy state by the addition of a photon. In chemiluminescence, the electron is in a high-energy state due to the creation of an unstable intermediate using a chemical reaction. Light is released when the intermediate breaks down into the final products of the reaction. Generally speaking, a fluorescence immunoassay is commonly faster, simpler, and cheaper to develop, while a chemiluminescence immunoassay could be more sensitive (from 10 to 100 times) and less susceptible to interference [26,49].

Norman et al. [34] assessed the feasibility, acceptability, and diagnostic effectiveness of a POCT troponin I assay (Abbott i-STAT, Abbott Laboratories, Abbott Park, IL, USA) to identify patients at low risk of AMI, to avoid unnecessary patient transfer to hospital and allow early discharge home in 12 rural general nursing practices (i.e., Pinnacle Midlands Health Network, Hamilton, New Zealand). A total of 180 patients (mean age 52 years, 88 women) were enrolled in this prospective observational pilot evaluation. After the clinical evaluation, 111 patients (61.7%) were considered to be at low risk and all were managed in rural general practice with no 30-day MACE (0%, 95% CI 0.0% to 3.3%). Of the 56 patients classified as non-low-risk and referred to hospital, 9 (16.1%) had a 30-day MACE [34]. Further, 13 non-low-risk patients were not transferred to hospital, with no events. Conversely, 94% of the low-risk patients reported good to excellent satisfaction with care. A good agreement was observed between the results found by POCT method and the hs-cTnT assay (Elecsys hs-cTnT method, Roche Diagnostics, Minato, Tokyo) performed in the hospital laboratory [34]. The authors concluded that the use of an accelerated diagnostic chest pain pathway incorporating the POCT troponin assay in a rural general practice setting was feasible and acceptable, and also effectively reduce the urgent transfer of low-risk patients to hospital [34].

Dawson et al. [72] analyzed the cost/benefit ratio of the pre-hospital testing using some POCT troponin methods. A total of 188,551 patients attended by ambulance for chest pain (mean age 61.9 years; 50.5% female) were enrolled in this economic evaluation based on cost-minimization analysis with the aim of evaluating the cost/benefit ratio linked to ambulance, emergency, and hospital attendance in the state of Victoria, Australia, between 1 January 2015, and 30 June 2019. The pre-hospital implementation of the POCT assay together with paramedic risk stratification for patients with acute chest pain could result in substantial cost savings. 

In the last 5 years, three POCT hs-cTnI methods have become commercially available worldwide [25,26,36,37]. In Table 1, the instrument characteristics and analytical performances of these POCT hs-cTnI methods are reported. Moreover, the clinical results of patients with thoracic pain or suspected ACS, using hs-cTnI POCT methods, have been published [27,28,30,31,33,73], as summarized in Table 2. 

Some recent documents and guidelines recommend the use of hs-cTnI POCT methods to rule out NSTEMI in the ED, using the entry sample or the rapid 0/1 h algorithm [25,26,36,37,74]. This approach should allow the rapid identification of low-risk patients who do not require further invasive tests and can be safely and quickly discharged, decreasing the overcrowding of the ED and the costs of care [25,26,36,37,74]. A very recent study [73] aimed to evaluate the diagnostic performance of an hs-cTnI POCT method (Atellica^®^ IM High-Sensitivity Troponin I (hs-cTnI) assay, Siemens Diagnostics, Marburg, Germany) for the rapid rule-out of AMI, using a single hs-cTnI measurement at presentation in patients presenting to an ED. Authors enrolled 1171 patients, including 97 (8.3%) with AMI, 78.3% of which were type 2 AMI (see also Table 2) [73]. The optimal rule-out value for the hs-cTnI POCT method threshold was <10 ng/L, which identified 519 (44.3%) patients as low-risk, with a sensitivity of 99.0% (95% CI, 94.4–100) and NPV of 99.8% (95% CI, 98.9–100) [73]. For type 1 AMI, sensitivity was 100% (95% CI, 83.9–100) and NPV 100% (95% CI, 99.3–100). Regarding myocardial injury, the sensitivity and NPV were 99.5% (95% CI, 97.9–100) and 99.8% (95% CI, 98.9–100), respectively [73]. For 30-day adverse events, sensitivity was 96.8% (95% CI, 94.3–98.4) and NPV 97.9% (95% CI, 96.2–98.9). Therefore, a single measurement strategy using hs-cTnI POCT methods enables the rapid identification of patients at low risk of AMI and 30-day adverse events, allowing potential early discharge after ED admission [73] (Figure 2).

In 2023, the IFCC Committee on Clinical Applications of Cardiac Bio-Markers (IFCC C-CB) [74] provided some practical educational pathways and technical information on the analytical characteristics and clinical relevance of POCT methods for cardiac biomarkers. A document reports some specific recommendations on the most appropriate use, the analytical performance, and results of POCT methods for cTn assay, based on the most recent clinical studies. Overall, POCT methods should be employed to diagnose NSTEMI, both in the ED and in other clinical settings, because they improve diagnostic efficacy and reduce TAT and waiting for patients in the ED [25,26,36,37,74]. This goal can be achieved by implementing a diagnostic pathway that includes the hs-cTnI POCT methods using 0–1 h rapid diagnostic algorithms, because patients at low risk of NSTEMI can be identified more quickly [25,26,36,37,74].

The implementation of hs-cTnI POCT methods requires not only a careful education of the personnel responsible for biomarker measurement, but also an accurate evaluation of analytical, clinical, and organizational issues related to the routine use of the POCT methods in the ED [25,26,36,37,74]. Particular attention should be paid to the pre-analytical and analytical phases due to the possible presence of analytical interference and also to the clinical difficulties related to cut-off values specific for age, sex, and method [14,16,18,19,24,25,26,36,37,57,74]. Moreover, there is a need for a validated system in order to guarantee a suitable laboratory information system (LIS), quality control of procedures, and clinical results obtained with hs-cTnI POCT methods in EDs [25,26,36,37,74].

### 3.4. Clinical Role of Cardiac-Specific Biomarkers in Patients with AHF Admitted to the ED

HF represents a relevant public health burden in European and North American countries, considering the high risk of morbidity and mortality, particularly among patients ≥ 65 years [56,75,76]. In Europe, the incidence of HF is currently about 3/1000 person-years (all age groups) or about 5/1000 person-years in adults [56]. Recent studies indicate that about 6 million US citizens and as many as 15 million Europeans are currently diagnosed with HF [75,76,77]. Moreover, due to the combined effect of population aging and improved survival from cardiovascular diseases, a further increase in the overall prevalence of HF is expected [56,75,76]. 

The 2015 consensus paper from the Heart Failure Association of the European Society of Cardiology, the European Society of Emergency Medicine, and the Society of Academic Emergency Medicine Acute Heart Failure has defined AHF as the new onset or worsening of symptoms and signs of HF with associated elevated circulating levels of natriuretic peptides [78]. AHF is considered the most frequent cause of unplanned hospital admission in patients aged >65 years, requiring immediate medical attention and urgent hospital admission [76,77,78,79,80]. Accordingly, emergency physicians play a pivotal role in the management of patients with AHF [76,77,78,79,80]. Diagnostic and therapeutic approaches of patients admitted to the ED with AHF significantly affect hospital length of stay, morbidity, and mortality, thereby inducing a direct impact on health and social costs [76,77,78,79,80].

Several recent studies and two meta-analyses confirmed the relevance of the measurement of cardiac-specific biomarkers in patients presenting to the ED with acute dyspnea or AHF [81,82,83,84,85,86,87,88,89,90,91,92,93,94,95]. In particular, cardiac NPs have already been included in the initial standard ED diagnostic workup of patients with dyspnea [76]. Specific cut-off values for BNP (i.e., 100 ng/ L) and NT-proBNP (i.e., 300 ng/L) have been recommended by some international guidelines for the diagnosis of AHF in patients admitted to the ED [56,76,95]. In addition to supporting the diagnosis of AHF, elevated NP concentrations are also useful for prognostication [56,76,95]. However, it should be noted that there are several cardiac and non-cardiac clinical conditions that may affect the diagnostic accuracy of the NPs assay, such as atrial fibrillation, elderly age, obesity, renal failure, and sepsis [44,54,56,77,96,97,98,99]. Due to the high intra-individual biological variation in NPs (especially for BNP), increments in measured biomarker levels ≥ 50% are required to indicate an exacerbation of HF [44,51,55,97,99].

The most important limitation of the clinical studies concerning the role of cardiac NPs (especially BNP and NT-proBNP) in patients admitted to the ED with AHF is that these studies have commonly enrolled relatively small, selected patient cohorts. Specific clinical characteristics (i.e., older age, renal disease, or obesity) are prevalent in AHF patients [56,76,77,78,79,80]. Accordingly, statistical modeling approaches that are able to consider some individual clinical characteristics of AHF patients may provide more consistent diagnostic performance across patient subgroups.

In 2022, a meta-analysis evaluated the NT-proBNP concentrations of 10,368 individual patients with AHF, with the purpose to develop and validate a decision support tool combining biomarker concentrations and clinical characteristics [91]. In this meta-analysis, the results of 14 studies from 13 countries (including randomized controlled trials and prospective observational studies) were evaluated [91]. The diagnostic performance of NT-proBNP threshold values in AHF patients can significantly vary due to comorbidities or pathophysiological variables [56,76,77,78,79,80]. Accordingly, it is better to evaluate the probability of AHF for individual patients, considering the NT-proBNP as a continuous clinical variable, using appropriate statistical models to take into account the possible disturbing effects of some pathophysiological variables and comorbidities [91]. 

In patients admitted to the ED with suspected ACS, including those with AHF, the measurement of hs-cTnI and hs-cTnT should be considered the first-line biomarkers, in order to identify the presence of myocardial injury and to quickly rule out ACS, leading to a significant reduction in the duration and costs of hospitalization [14,15,16,17,18,19,20,56,76,77,88,100,101,102]. Furthermore, several recent clinical studied have demonstrated that hs-cTnI and hs-cTnT are useful predictor biomarkers for mortality and MACE, even in the setting of AHF [17,19,22,53,84,102,103,104]. In particular, according to the Fourth Universal Definition of Myocardial Infarction [8], an increase in hs-cTnI and hs-cTnT above the 99th percentile URL value proves the presence of myocardial injury in AHF patients. For the diagnosis of myocardial injury in AHF patients, it is important to remember that cut-off (i.e., 99th percentile URL) values specific for method and sex should be preferred for hs-cTnI methods [8,14,16,18,19,39,40,41,49,57,58].

According to the most recent clinical studies, high levels of both cardiac-specific biomarkers (i.e., NPs and hs-cTn) in AHF patients are consistently associated with an increased risk of morbidity and mortality, due to the combined presence of a severe reduction in cardiac function and myocardial injury (or even ACS) [56,76,77,80,87,88,100,101,102,104]. The relevant effect on mortality and MACE in patients admitted to the ED with AHF for several days (or even months), associated with the progressive increase in both specific cardiac biomarkers, has recently been well illustrated in the particular clinical conditions related to COVID-19 infection [105,106,107,108,109,110]. In particular, Sandoval et al. [6] stated that the assay of specific cardiac biomarkers (especially hs-cTnI and hs-cTnT) in serial samples collected from patients admitted to the ED and hospitalized for COVID-19 facilitates risk stratification, helps decide when to use cardiac imaging, and indicates stage categorization and disease phenotyping among patients.

## 4. Conclusions

In order to balance the risk between a hasty discharge versus the potential harms caused by a cardiac assessment in patients admitted to the ED with suspected acute cardiovascular disease, the recent expert documents and guidelines recommend the measurement of specific cardiac biomarkers (NPs and hs-cTn) for the early identification of the presence of myocardial dysfunction and/or injury and to significantly reduce the duration and costs of hospitalization [8,10,14,15,16,17,18,19,20,56,76,77,98,100,101]. Moreover, specific cardiac biomarkers (especially hs-cTnI and hs-cTnT) are useful predictors of mortality and MACE in patients admitted to the ED with suspected acute cardiovascular disease [17,19,22,53,84,102,103,104].

The very recent development of some POCT methods capable of measuring hs-cTnI (as well as NPs) represents a fundamental advance because these methods can significantly reduce the TAT of biomarker measurement [25,26,36,37]. Notably, POCT methods can be used at home, in an outpatient clinic (primary care) or in an ambulance, in order to reduce unnecessary transfer to hospital and allow patient management in rural general practice [14,34,35,36,37,72]. Furthermore, some POCT methods have the advantages of requiring a smaller amount of blood for the analysis compared to the methods used by automated platforms in clinical laboratories [26,36,37,49]. Only a drop of whole blood collected with a puncture with a needle from the tip of the finger in very debilitated patients, or from the heel of the foot in neonates is sufficient for the assay of cardiac biomarkers (i.e., BNP/NT-proBNP and hs-cTnI) with some POCT methods [26,36,37,49]. The possibility to perform the biomarker assay using a drop of whole blood is essential for gaining acceptance into practice in emergency medicine and other decentralized settings [26,36,37,49].

To guide the implementation of the POCT methods, capable of measuring both the specific cardiac biomarkers, into routine clinical practice, clinical scientific societies and laboratory medicine societies should promote collaborative studies specifically designed for the evaluation of the analytical performance and, especially, the cost/benefit ratio resulting from the use of the rapid algorithms for the diagnosis of NSTEMI and the use of POCT methods in the ED clinical practice.

## Figures and Tables

**Figure 1 ijms-24-15998-f001:**
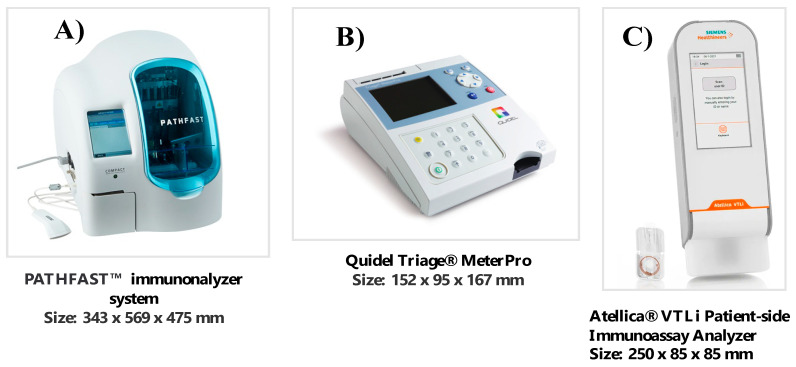
The three hs-cTnI POCT methods now on the market, with their specific dimensions: (**A**) like a benchtop instrument, (**B**) like a desk phone; (**C**) like a smartphone. POCT: Point-of-care testing.

**Figure 2 ijms-24-15998-f002:**
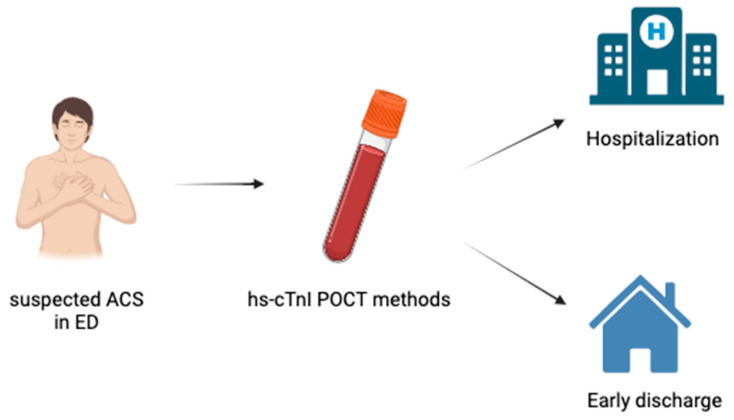
hs-cTnI POCT methods enable a rapid identification of patients at low risk of AMI and 30-day adverse events, allowing potential early discharge after ED admission. POCT: Point-of-care testing; AMI: acute myocardial infarction; ED: Emergency department.

**Table 1 ijms-24-15998-t001:** Instrument characteristics and analytical performance of commercially available hs-cTnI POCT methods.

hs-cTnI POCT Method	Instrument Characteristics	LoD (ng/L)	LoQ 10% (ng/L)	99th Percentile URL (ng/L)	Reference
PATHFAST™ POC hs-cTnI (PHC Europe B.V., Nijverheidsweg, The Netherlands).	This method combines the technologies of chemiluminescence for signal detection (enzyme alkaline phosphatase bound to anti-cTnI monclonal antibodies) and magnetic migration for the separation of the bound phase, using antibodies marked by magnetic particles.	2.9	11.0	W 21.1 (13.4–25.3) * M 27.0 (18.5–27.7) *	Sorensen NA et al. [27]
Quidel TriageTrue High Sensitivity Troponin I Test (Quidel Corporation Headquarters, San Diego, CA 92121, USA).	The immunofluorescent Quidel TriageTrue method uses a few drops of whole blood or EDTA plasma for biomarker measurement with the Triage^®^ MeterPro instrument, which has the weight of a cell phone.	0.7–1.6	4.4–8.4	W 14.4 (13.1–28.7) ** M 25.7 (18.3–37.6) **	Boeddinghaus J et al. [28]
Siemens POC Atellica ^®^ VTLi hs-cTnI (Siemens Healthineers, Erlangen, Germany).	This method uses Magnotech^®^-type biosensors to separate the cTnI fraction, bound by antibodies to magnetic beads, from the free fraction and detects the signal using the imaging technique called Frustrated Total Internal Reflection (FTIR). This method uses the Atellica^®^ VTLi Patient-side Immunoassay Analyzer instrument, which has the dimensions of a hand-held instrument.	1.2	6.7	W 18.0 (9.0–78.0) ** M 27.0 (21.0–37.0) **	Apple FS et al. [29]

LoD: limit of detection; LoQ 10% CV: limit of quantitation 10% CV; W: women; M: men. * 95% confidence interval; ** 90% confidence interval.

**Table 2 ijms-24-15998-t002:** Results reported in clinical studies related to hs-cTnI POCT methods.

Study [Reference]	Study Type	N° and Type of Population	POCT Method	Main Clinical Results	Conclusions
Sörensen NA, et al. 2019 [27]	prospective cohort study (enrolment time: from July 2013 to July 2016)	669 patients presenting to the ED with suspected MI (STEMI excluded).	POCT hs-cTnI assay (PATHFAST hs-cTnI)	Negative predictive value of 99.7% (95% CI, 98.1–100.0%) and 48.0% of patients ruled out, whereas 14.6% were ruled in with a positive predictive value of 86.5% (95% CI, 77.6–92.8%).	The diagnostic performance of the new POCT assay was highly comparable to that of the laboratory hs-cTnI methods
Boeddinghaus J, et al. 2020 [28]	prospective international multicenter study (12 centers, 5 countries)	1261 adult patients presenting to the ED with symptoms suggestive of MI with an onset or peak within the last 12 h.	POCT hs-cTnI (TriageTrue) assay	The 0/1 h algorithm ruled out 55% of patients (NPV: 100%; 95% CI: 98.8% to 100%), and ruled in 18% of patients (PPV: 76.8%; 95% CI: 67.2% to 84.7%).	The POCT hs-cTnI assay provides high diagnostic accuracy in patients with suspected AMI with a clinical performance that is at least comparable to that of best-validated central laboratory assays.
Apple FS, et al. 2022 [30]	2 prospective observational studies	1086 patients (8.1% with MI) from a US derivation cohort (SEIGE) and 1486 (5.5% MI) from an Australian validation cohort (SAMIE). All of these patients presented to the ED with suspected acute coronary syndrome.	Whole-blood POC hs-cTnI assay	A derivation whole-blood POC hs-cTnI provided a sensitivity of 98.9% (95% CI, 93.8–100%) and negative predictive value of 99.5% (95% CI, 97.2–100%) for ruling out MI. In the validation cohort, the clinical sensitivity was 98.8% (95% CI, 93.3–100%), and negative predictive value was 99.8% (95% CI, 99.1–100%); 17.8% and 41.8%, respectively, were defined as low risk for discharge. The 30-day adverse cardiac events were 0.1% (*n* = 1) for SEIGE and 0.8% (*n* = 5) for SAMIE.	A POC whole-blood hs-cTnI assay permits accessible, rapid, and safe exclusion of MI and may expedite discharge from the emergency department.
Bruinen AL, et al. 2022 [31]	prospective, observational cohort study (enrolled from September 2019 until November 2020)	152 adult patients (55% female, 45% male) referred to the cardiac ED because of acute chest pain suspected for ACS.	Atellica VTLi Patient-side Immunoassay Analyzer (method and sample comparison) using different sample types including capillary blood, in comparison with standard laboratory hs-cTnI testing	No significant difference was observed between venous whole blood vs. plasma analyzed. The difference between capillary blood and venous blood showed a constant bias of 7.1%, for which a correction factor was implemented.	No clinically relevant differences were observed for the capillary POC results compared to plasma analyzed with a standard immunoassay analyzer.
Gunsolos IL, et al. 2022 [33]	study cohort	1089 patients (418 F and 671 M) presenting to ED, with suspected AMI (excluded: age < 21 years, pregnancy, trauma, transferred from an outside hospital).	Whole-blood, point-of-care (POC) high-sensitivity cardiac troponin I (hs-cTnI) assay (Siemens Atellica VTLi)	At baseline (0 h), POC hs-cTnI assay had a sensitivity of 65.7% (95% CI 47.8–80.9) for females and 67.9% (54.0–79.7) for males and NPV of 96.4% (93.9–98.1) for females and 96.7% (94.9–98.0) for males. At 2 h, sensitivity improved to 82.9% (66.4–93.4) for females and 80.4% (67.6–89.8) for males, while NPV improved to 98.2% (96.1–99.3) and 97.9% (96.3–99.0), respectively.	For central laboratory assays (ARCHITECT and Atellica hs-cTnI methods), comparable diagnostics were observed at 2 h.
Fabre-Estremera, et al. 2023 [73]	prospective, observationalcohort study	1171 patients (mean age of 58.9 years and 38.2% female) presenting to an US ED.	Atellica ^®^IM High-Sensitivity Troponin I assay	AMI occurred in 97 patients (8.3%), 78.3% of which were type 2 AMI. The optimal rule-out POCT hs-cTnI method threshold was <10 ng/L, which identified 519 (44.3%) patients as low-risk at presentation, with sensitivity of 99.0% (95% CI, 94.4–100) and NPV of 99.8% (95% CI, 98.9–100). For type 1 AMI, sensitivity was 100% (95% CI, 83.9–100) and NPV 100% (95% CI, 99.3–100). Regarding myocardial injury, the sensitivity and NPV were 99.5% (95% CI, 97.9–100) and 99.8% (95% CI, 98.9–100), respectively. For 30-day adverse events, sensitivity was 96.8% (95% CI, 94.3–98.4) and NPV 97.9% (95% CI, 96.2–98.9).	A single-measurement strategy using the POCT hs-cTnI method I able to rapidly identify patients at low risk of AMI and 30-day adverse events, allowing potential early discharge after ED presentation.

## Data Availability

No new data were created or analyzed in this study. Data sharing is not applicable to this article.

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
