# Peer review of "Measurement of Cardiac-Specific Biomarkers in the Emergency Department: New Insight in Risk Evaluation"

_ijms, 2023, doi:10.3390/ijms242115998_

Round 1
Reviewer 1 Report
Comments and Suggestions for Authors
The Review entitled „Measurement of cardiac-specific biomarkers in the Emergency Department: new insight in risk evaluation" is of interest. However, the reviewt needs to be improved.
- Clinical issue, the authors stated:
"To analyze some models and clinical issues related to the implementation of accelerated diagnostic protocols in …… in patients admitted to the Emergency Departmen (ED)“. However, diagnostic models shown here are only few and not enough to reflect the procedure in the ED in daily basis manner. The review is limited only to two assays (hs-cTnI and hs-cTnT), there is no information on risk-stratification, ST-elevation, or other biomarker panels (CK and CK-MB, CRP, etc…), why?
Review design: Table 1, information mostely cited from literature are limited and not impressive. Why is it important to add in „instrument characteristics“, if the system detect chmiluminescence or immunofluorescence?? What is advantages and disadvantages of the different methods?? Same to table 2, difficult to read, main clinical results, which were not meaningful results.
Comments on the Quality of English LanguagePlease avoid repeating many abbrevations and vice versa. Revise the whole manuscript and take attention to numerous of typos errors.
. Tables were not easy to interpret and read.
Author Response
Dear reviewer,
thank you very much for your time.
We really appreciated the comments and tried to implement the manuscript, following the suggestions provided.
In detail:
- Table 1. The following paragraph has been added to the revised version, according to the suggestion made by the Reviewer (page 5 line 227). “ Form an analytical point of view, chemiluminescence and fluorescence techniques are used interchangeably in POCT hs-cTnI methods (Table 1) [26, 49]. Both these two techniques give off a photon as an electron relaxes from a higher energy state to a lower energy state, but the difference lies in the method used to excite that electron to a higher energy state in the first place. In fluorescence the electron is kicked up to a higher energy state by the addition of a photon. In chemiluminescence the electron is in a high-energy state due to the creation of an unstable intermediate using a chemical reaction. Light is released when the intermediate breaks down into the final products of the reaction. Generally speaking, a fluorescence immunoassay is commonly more fast, simple and cheap to develop, while a chemiluminescence immunoassay could be more sensitive (from 10 to 100 times) and less susceptible to interferences [26, 49].”
- Table 2. We modified the Table 2.
-We corrected typographical errors throughout the manuscript.
We hope that with these changes the manuscript will now be more suitable for publication.
Thank you again
Reviewer 2 Report
Comments and Suggestions for Authors
The article entitled "Measurement of cardiac-specific biomarkers in the Emergency Department: new insight in risk evaluation" written by Aspromonte and colleagues is a review article describing the usefulness of troponin and atrial natriuretic peptide measurement in the stratification of patients at risk of heart failure presenting to the hospital emergency department.
1. In the abstract, indicate the significance of the biomarkers, if possible.
2. Section 3, titled Discussion (line 114), should be renamed.
3. Correct references according to the author guidelines, e.g., reference 2 (line 52) should appear after references 3-8 (line 50). Reference 56 (line 179) is written after references 48-51 (line 163), but reference 52 appears up to line 220.
4. References 81-95 (lines 327-328) should appear in square brackets.
5. References 71, 103, and 104 of the references are not found in the manuscript.
6. In Tables 1 and 2, I recommend citing only the references, e.g.: [27], and also replace the F of female with W to homogenize with respect to the rest of the labels used. Also include the volume of blood or serum used for each detection method.
7. Update reference 78. For example: https://doi.org/10.1002/ejhf.2333.
8. I suggest the authors include a table with reference values for troponins and atrial natriuretic peptides and values indicating a high risk of acute myocardial infarction with POCT testing.
9. The authors should include information on the advantages and disadvantages of POCT methods, for example, sensitivity and specificity compared with conventional laboratory tests.
Comments on the Quality of English LanguageCorrect typographical errors throughout the manuscript, e.g., extra spaces between words, the word "authoers" in the last paragraph of Table 2.
Author Response
Dear reviewer,
thank you very much for your time.
We really appreciated the comments and tried to implement the manuscript, following the suggestions provided.
In detail:
- We renamed the Section 3, “Cardiac biomarkers in the setting of ED”.
- We corrected the order of references
- We corrected references 81-95 in square brackets.
- We verified that references 71 (line 212), 103 and 104 (line 372, included in 102-105) are found in the manuscript.
- We revised the Tables 1 and 2, citing only the references, and also replacing the F of female with W.
- We updated reference 78.
- As reported many times in the manuscript, there are very large differences among the cut-off values of assay methods of natriuretic peptides (BNP and NT-proBNP) as well hs-cTnI and hs-cTnT. Accordingly, it is impossible to report in one or even two tables all the data requested by the Reviewer. However, the cut-off values for myocardial injury of POCT hs-cTnI methods (i.e., the 99th percentile URL values) are reported in Table 1 and the results of the diagnostic accuracy of the clinical studies with the relative cut-values are reported in Table 2. Several articles, cited in the manuscript, already reported the cut-off values of hs-cTnI and hs-cTnT methods [14-20, 24, 36-40,49, 57,74]. It is important to note that there are more than 10 different hs-cTnI methods commercially available in European and North America countries and almost two different versions of the hs-cTnT method (Roche Diagnostics), and two different hs-cTnI methods from Cinese manufacturers.
- Authors already reported in the Conclusions part of the manuscript the most important clinical advantages of POCT hs-cTnI methods (page 11, lines 390-394). However, according to the suggestions made by the Reviewer, the following sentences have been added in the revised version of the manuscript (page 11, lines 406– 414). “Furthermore, some POCT methods have the advantages of requiring a smaller amount of blood for the analysis compared to the methods used by automated platforms in the clinical laboratories [26,36, 37,49]. Only a drop of whole blood collected with a puncture with a needle from the tip of the finger in very debilitated patients, or from the heel of the foot in neonates is sufficient for the assay of cardiac biomarkers (i.e., BNP/NT-proBNP and hs-cTnI) with some POCT methods [26,36,37,49]. The possibility to perform the biomarker assay using a drop of whole blood is essential for gaining acceptance into practice in emergency medicine and other decentralized settings [26,36,37,49].
-We corrected typographical errors throughout the manuscript.
We hope that with these changes the manuscript will now be more suitable for publication.
Thank you again
Round 2
Reviewer 2 Report
Comments and Suggestions for Authors
Just check the order of references in the manuscript
Author Response
Dear reviewer,
thank you again for your time.
We have checked the references and added two figures, according to the editor's instructions.